# "I am afraid of being treated badly if I show it": A cross-sectional study of healthcare accessibility and Autism Health Passports among UK Autistic adults

**Aimee Grant**[1]*, **Sarah Turner**[2], **Sebastian C. K. Shaw**[3], **Kathryn Williams**[4,5], **Hayley Morgan**[1], **Rebecca Ellis**[1], **Amy Brown**[1]

**1** School of Health and Social Care, Swansea University, Swansea, United Kingdom, **2** Children's Hospital Research Institute, University of Manitoba, Winnipeg City, Canada, **3** Department of Medical Education, Brighton and Sussex Medical School, Brighton and Hove, United Kingdom, **4** Autistic UK CIC, Polegate, United Kingdom, **5** School of Social Sciences, Cardiff University, Cardiff, United Kingdom

* aimee.grant@swansea.ac.uk

## Abstract

### Background

Autistic people are more likely to experience stigma, communication barriers and anxiety during healthcare. Autism Health Passports (AHPs) are a communication tool that aim to provide information about healthcare needs in a standardised way. They are recommended in research and policy to improve healthcare quality.

### Aim

To explore views and experiences of AHPs among Autistic people from the UK who have been pregnant.

### Methods

We developed an online survey using a combination of open and closed questions focused on healthcare impairments and views and experiences of AHPs. Data were anlaysed using descriptive statistics, Kruskal-Wallis tests, and content analysis.

### Findings

Of 193 Autistic respondents (54% diagnosed, 22% undergoing diagnosis and 24% self-identifying), over 80% reported anxiety and masking during healthcare always or most of the time. Some significant differences were identified in healthcare (in)accessibility by diagnostic status. Only 4% of participants knew a lot about AHPs, with 1.5% of participants using one at least half of the time. Almost three quarters of respondents had not previously seen an AHP. Open text responses indicated that the biggest barrier to using an AHP was a belief that health professionals would discriminate against Autistic patients. Additional barriers

the respondents of our survey to share the de-identified dataset with the general public (Ethics committee: Swansea University's School of Health and Social Care; application reference: 280122). Data requests can be directed to the data access committee of our Autism research group (contact via AutismM2M@swansea.ac.uk).

**Funding:** AG, KW, AB and EH received funding for this research from the Swansea University Accelerate Health Tech Centre. Reference: 07/09/21. HM and RE's time was funded by this grant. The funders had no role in study design, data collection and analysis, decision to publish, or preparation of the manuscript.

**Competing interests:** I have read the journal's policy and the authors of this manuscript have the following competing interests: AG is a non-executive Director of Disability Wales. SS is the research lead for Autistic Doctors International, a peer support and advocacy group which is not a legal entity. KW is a non-executive director for Autistic UK a Community Interest Company. This does not alter our adherence to PLOS ONE policies on sharing data and materials. None of the three individuals receive a salary for these affiliations. The organisations did not have any role in the study design, data collection and analysis, decision to publish, or preparation of the manuscript. The remaining authors declare no competing interests.

included staff lack of familiarity with AHPs and respondents expecting a negative response to producing an AHP.

## Conclusions

Our findings suggest that AHPs are not reducing health inequalities for Autistic adults who have been pregnant. Alternative solutions are needed to reduce health inequalities for Autistic people.

## Background

Autism is a part of cognitive diversity, and is increasingly recognised as a form of difference which may be advantageous within an accessible environment [1]. However, Autistic people are routinely disadvantaged by an ableist society [2] designed to meet the needs of neurotypical people, and which does not respect Autistic differences as viable alternatives [3]. Autistic people have worse physical and mental health and a shorter life expectancy [4], which may be due to healthcare inaccessibility [5]. This includes systematic issues, such as discrimination from health professionals [6], communication differences [7], and lack of recognition of Autistic differences in experiencing interoception (internal bodily signals) [8] and pain [9]. Barriers to accessing healthcare have also been associated with negative health outcomes for Autistic people [10]. A decade ago, Autistic adults' experiences of health and healthcare were underexplored [11], but there has recently been significant growth in this research area. For example, in relation to Magnetic Resonance Imaging a body of work includes a systematic review [12] and primary research [13] leading to recommendations for clinical practice [14].

Alongside general challenges in accessing healthcare, Autistic people who are Assigned Female At Birth (AFAB) have been historically underdiagnosed, although this is beginning to improve with diagnosis of people AFAB during adulthood [15]. Exploration of routine data has highlighted that Autistic adults AFAB have worse health compared to non-Autistic peers, including increased chronic physical and mental ill health, history of assault and material deprivation [16]. Increased use of healthcare surveillance occurs during pregnancy, and it may be a particularly difficult time for Autistic people due to increased sensory challenges [17]. Furthermore, Autistic people have an increased risk of pre-term labour and preeclampsia [18] and face additional barriers to breastfeeding [19, 20].

Autism Health Passports (AHPs) are a range of communication tools that aim to provide information about healthcare needs in a standardised way. The English Autism Act 2021 and numerous UK policy documents [21–23] and clinical guidelines [24] recommend AHPs. This is also found in international literature, where it is suggested AHPs can reduce health inequalities [25]. The UK National Autistic Society [26] and some UK NHS Trusts have produced AHP templates. However, our realist review has identified that they are under-theorised, under-evaluated and there is limited likelihood of any AHP-related positive outcomes [27].

## Methodology

### Aim

To use an online cross-sectional survey to explore barriers to healthcare access and AHP views and experiences of Autistic adults from the UK who had been pregnant.

We follow the CROSS guidance on reporting of survey studies [28].

## Community involvement and reflexivity

Five of the researchers, including the principal investigator, (AG, SS, KW, HM and RE) are Autistic. AG is a multiply Disabled gender ambivalent white person. She is aged 41, and has a PhD in social policy. ST is a white female, aged 32 and mother to one. She has a Masters in Community Health. SS is a 31-year-old, white, cis-gendered, autistic man. He works as a doctor in the UK NHS and is also a Lecturer in Medical Education. His PhD focused on the experiences of neurodivergent medical students and doctors. KW is an Autistic and ADHD white woman. She is 38 years old, has an MSc in Social Science Research Methods, and is undertaking a PhD in Social Policy. Furthermore, KW is the research director of Autistic UK, an Autistic-led organisation which aims to improve the lives of Autistic people. HM is a late-diagnosed Autistic white woman who is 37 and is a disabled carer. She has an MSc in Autism and Related Conditions and is currently undertaking a PhD in health humanities. RE is a 32-year-old white Autistic woman with a PhD in Human and Health Sciences. AB is a 42-year-old white woman. She has a PhD in health psychology. We also sought feedback on our research design from Autistic people with experience of pregnancy via relevant Facebook groups and through Twitter, receiving feedback from 27 people. The use of gender neutral and identity-first language was strongly preferred and, as such is used throughout the manuscript.

## Survey design

The survey was split into four parts: demographics (Table 1), healthcare experiences (Table 2), AHP views and experiences (Table 4), and maternity experiences which are reported separately [20]. A mixture of open and closed questions was used, including an open text box at the end of each Part for additional information. A five-point Likert scale with a sixth option of 'prefer not to say' was used for many of the closed questions, with scales ranging from "always" to "never" and "strongly agree" to "strongly disagree". The survey was not subjected to pretesting, but Autistic people were involved in the design of the measures. Participants were told to contact AG if they had any concerns when completing the survey, and that they did not have to answer any questions that they did not want to. This paper draws on data from Parts 1–3. The questionnaire is available as S1 Appendix.

## Participants and eligibility criteria

The online survey, hosted on *Qualtrics*, was open from 10th February to 31st March 2022. Participants were required to be:

- Autistic, including those diagnosed, undergoing diagnosis and those who self-identified as Autistic, in recognition of barriers to diagnosis for those AFAB in the UK,

- Aged >18 years,

- Living in the UK,

- Pregnant, or to have been pregnant previously, including those who had only experienced pregnancy loss.

Multiple responses from the same IP address were restricted by the *Qualtrics* platform; the same functionality allowed participants to return to their survey for up to a week if they had not submitted their results. The survey utilised a convenience sample and was advertised through the social media networks of the researchers, including Autistic UK, the Autistic-led organisation that KW represents, and through two Facebook groups for Autistic people who (i) breastfeed, and (ii) are planning to become pregnant or are pregnant, and parents. We did

**Table 1. Demographic profile of sample.**

| Demographic | Sub-categories | Total (n = 193) | Autism diagnosis status | | | |
|---|---|---|---|---|---|---|
| | | | Diagnosed (n = 104, 53.9%) | Undergoing diagnosis (n = 42, 21.8%) | Self-identifying (n = 43, 22.3%) | Other (n = 3, 1.5%) |
| **Autism Diagnosis status** | Diagnosed | 104 (53.9%) | - | - | - | - |
| | Undergoing diagnosis | 42 (21.8%) | - | - | - | - |
| | Self-identify | 43 (22.3%) | - | - | - | - |
| | Other | 3 (1.6%) | - | - | - | - |
| | Missing | 1 (0.5%) | - | - | - | - |
| **Communication preference** | Speaking | 164 (85.0%) | 89 (85.6%) | 37 (88.1%) | 37 (86.0%) | 1 (33.3%) |
| | Sign language | 2 (1.0%) | 1 (1.0%) | - | 1 (2.3%) | - |
| | Assistive and Augmentative communication | 1 (0.5%) | 1 (1.0%) | - | - | - |
| | Other | 23 (11.9%) | 12 (11.5%) | 5 (11.9%) | 4 (9.3%) | 2 (66.7%) |
| | Missing | 3 (1.6%) | | | | - |
| **Age** | Mean | 36.5 | 36.1 | 35.8 | 37.5 | 45.67 |
| | Range | 19–63 | 19–57 | 24–63 | 24–54 | 37–60 |
| **Gender identity** | Cis woman | 159 (82.4%) | 84 (80.8%) | 37 (88.1%) | 37 (86.0%) | 1 (33.3%) |
| | Intersex | 1 (0.5%) | - | 1 (2.4%) | - | - |
| | Trans man | - | - | - | - | - |
| | Non-binary | 16 (8.3%) | 9 (8.7%) | 3 (7.1%) | 4 (9.3%) | - |
| | Other | 14 (7.3%) | 10 (9.6%) | 1 (2.4%) | 1 (2.3%) | 2 (66.7%) |
| | Prefer not to say | 2 (1.0%) | 1 (1.0%) | - | 1 (2.3%) | - |
| | Missing | 1 (0.5%) | - | - | - | – |
| **Ethnicity** | White | 174 (90.2%) | 94 (90.4%) | 37 (88.1%) | 42 (97.7%) | 1 (33.3%) |
| | Mixed or multiple | 7 (3.6%) | 6 (5.8%) | 1 (2.4%) | - | - |
| | Asian or Asian British | 3 (1.6%) | 1 (1.0%) | 1 (2.4%) | 1 (2.3%) | - |
| | Black, African, Caribbean or Black British | - | - | - | - | - |
| | Other | 6 (3.1%) | 2 (1.9%) | 3 (7.1%) | - | 1 (33.3%) |
| | Prefer not to say | 2 (1.0%) | 1 (1.0%) | - | - | 1 (33.3%) |
| | Missing | 1 (0.5%) | - | - | - | - |
| **Disability other than being Autistic** | Yes | 138 (71.5%) | 81 (77.9%) | 28 (66.7%) | 26 (60.5%) | 3 (100%) |
| | No | 49 (25.4%) | 21 (20.2%) | 14 (33.3%) | 14 (32.6%) | - |
| | Prefer not to say | 5 (2.6%) | 2 (1.9%) | - | 3 (7.0%) | - |
| | Missing | 1 (0.5%) | - | - | - | - |
| **Disability impact on day-to-day activities** | A lot | 47 (24.4%) | 32 (30.8%) | 7 (16.7%) | 7 (16.3%) | 1 (33.3%) |
| | A little | 103 (53.4%) | 52 (50.0%) | 24 (57.1%) | 25 (58.1%) | 2 (66.7%) |
| | Not at all | 33 (17.1%) | 15 (14.4%) | 9 (21.4%) | 9 (20.9%) | - |
| | Prefer not to say | 4 (2.1%) | 1 (1.0%) | 2 (4.8%) | 1 (2.3%) | - |
| | Missing | 6 (3.1%) | - | - | - | - |
| **Highest qualification** | None | 2 (1.0%) | - | 2 (4.8%) | - | - |
| | GCSE | 16 (8.3%) | 10 (9.6%) | 3 (7.1%) | 3 (7.0%) | - |
| | A Levels | 20 (10.4%) | 13 (12.5%) | 4 (9.5%) | 3 (7.0%) | - |
| | NVQ | 25 (13.0%) | 17 (16.3%) | 4 (9.5%) | 4 (9.3%) | - |
| | Undergraduate degree | 61 (31.6%) | 30 (28.8%) | 11 (26.2%) | 20 (46.5%) | - |
| | Taught postgraduate | 47 (24.4%) | 26 (25.0%) | 11 (26.2%) | 9 (20.9%) | 1 (33.3%) |
| | Doctorate | 12 (6.2%) | 5 (4.8%) | 6 (14.3%) | 1 (2.3%) | - |
| | Other | 9 (4.7%) | 3 (2.9%) | 1 (2.4%) | 3 (7.0%) | 2 (66.7%) |
| | Missing | 1 (0.5%) | - | - | - | - |

*(Continued)*

**Table 1.** (Continued)

| Demographic | Sub-categories | Total (n = 193) | Autism diagnosis status | | | |
|---|---|---|---|---|---|---|
| | | | Diagnosed (n = 104, 53.9%) | Undergoing diagnosis (n = 42, 21.8%) | Self-identifying (n = 43, 22.3%) | Other (n = 3, 1.5%) |
| Location | England | 143 (74.1%) | 74 (71.2%) | 32 (76.2%) | 36 (83.7%) | 1 (33.3%) |
| | Scotland | 19 (9.8%) | 12 (11.5%) | 5 (11.9%) | 2 (4.7%) | - |
| | Wales | 24 (12.4%) | 16 (15.4%) | 4 (9.5%) | 2 (4.7%) | 2 (66.7%) |
| | Northern Ireland | 6 (3.1%) | 2 (1.9%) | 1 (2.4%) | 3 (7.0%) | - |
| | Missing | 1 (0.5%) | - | - | - | - |

not undertake a sample size calculation for this exploratory research. Participants were required to read a detailed information sheet before consenting to complete the survey in writing on the *Qualtrics* platform. The study received ethical approval from the School of Health and Social Care, Swansea University. Study materials, (the advert and participant information sheet) highlighted that participants would be given the opportunity to opt in to a prize draw, but were not required to do so to take part. When the survey closed, ten participants who opted into the prize draw were randomly selected to receive a £20 Amazon gift voucher. Personal information collected to facilitate entry into the 'prize draw' was removed from responses prior to analysis. Data was stored on password protected servers. All aspects of the study were performed in accordance with the ethical standards set out in the 1964 Declaration of Helsinki.

## Analysis

Respondents who completed any questions in addition to demographic data were included in the analysis. Quantitative data were analysed using descriptive statistics within IBM® SPSS®

**Table 2.** Experience of Autistic challenges in a healthcare context.

| | Always/Most of the time | | Half of the time | | Sometimes/never | |
|---|---|---|---|---|---|---|
| | N | % | N | % | N | % |
| **Agreement shows impairment** | | | | | | |
| Awareness of pain, injury, or discomfort (n = 192) | 128 | 66.7% | 24 | 12.5% | 40 | 20.8% |
| Delaying diagnostic healthcare appointments with recurrent / intermittent symptoms (n = 192) | 154 | 80.3% | 17 | 8.9% | 21 | 11.0% |
| Feeling anxious about telephoning healthcare services e.g.: to book appointments (n = 192) | 173 | 90.1% | 6 | 3.1% | 13 | 6.8% |
| Delaying making phone calls e.g.: to book appointments (n = 190) | 159 | 83.6% | 9 | 4.7% | 22 | 11.6% |
| Feeling anxious because of sensory experiences in waiting rooms (n = 192) | 130 | 67.7% | 22 | 11.5% | 40 | 20.9% |
| Experience frustration or misunderstandings when communicating with health professionals (n = 192) | 84 | 43.7% | 35 | 18.2% | 71 | 37.0% |
| Feeling anxious during healthcare appointments (n = 191) | 158 | 82.7% | 14 | 7.3% | 19 | 9.9% |
| Masking during healthcare appointments (n = 191) | 166 | 86.9% | 10 | 5.2% | 15 | 7.8% |
| Experience difficulty describing pain (n = 191) | 126 | 65.9% | 23 | 12.0% | 42 | 21.9% |
| Experience difficulty understanding health professional long/detailed communication (n = 190) | 104 | 54.8% | 24 | 12.6% | 62 | 32.6% |
| When distressed, communication skills are reduced (n = 191) | 151 | 79.0% | 21 | 11% | 19* | 10% |
| Experience difficulty due to sensory experiences in healthcare appointments (n = 191) | 118 | 61.7% | 27 | 14.1% | 46 | 24.1% |
| **Disagreement shows impairment** | | | | | | |
| Confident will be understood when describing physical symptoms (n = 191) | 29 | 15.1% | 37 | 19.4% | 125 | 65.4% |
| Find it easy to understand lengthy instructions (n = 191) | 41 | 21.5% | 32 | 16.8% | 118 | 61.8% |
| Managing to follow post-appointment instructions exactly (n = 191) | 89 | 46.5% | 35 | 18.3% | 67 | 35.1% |

*All 19 responses were "sometimes", with no "never" responses to this question

Statistics V28 by SS and AG overseen by AB; missing data were excluded. Kruskall-Wallis analyses were conducted to investigate whether there was a relationship between diagnostic status and healthcare accessibility. A mean "Autism Health Passport score" was constructed for each respondent who had answered at least six of the eight AHP questions, by adding the scale responses ("strongly agree" = a score of 1, up to "strongly disagree" = a score of 5) to each question on views of AHPs and dividing by the number of questions responded to (maximum of eight). An Analysis of Variance (ANOVA) was then used to explore any differences between this overall Autism Health Passport score and respondents' diagnostic status. Questions with open text responses were subjected to content analysis [29] by ST overseen by AG. This involved AG and ST familiarizing themselves with the data. ST then coded the data, including generating definitions. Codes were applied to either full responses to a question or to part of responses. Multiple codes were used were required. ST and AG had a series of four analysis meetings to refine the coding framework, including any latent meanings within responses, and to ensure shared understandings were reached. We used this as a quality assurance measure in place of double coding, which is sometimes included in content analysis.

## Results

### Demographics

193 participants were included in the analysis. Over half were formally diagnosed (53.9%), with a further fifth undergoing diagnosis (21.8%) (Table 1). "Other" responses included health professionals suggesting that the person was Autistic but no formal diagnosis; these were recoded as self-identifying for the quantitative analysis. Speaking was the most frequent form of communication (85.0%), with 13 of the "other" responses identifying writing as the participants' preference. Participants were asked if their communication preferences varied during times of stress via an open text box which received 114 responses. 94 participants noted a preference for using written communication during times of stress, for example: "At times of stress I always prefer to communicate via text messages/email/writing". Ten participants noted that they used sign language, including British Sign Language or Makaton during times of stress. 82.4% of the sample were cis-gendered women, with the "other" category comprising of eight people who were various forms of gender fluid; three who were agender; and three participants stating that they were "biologically female". The sample was largely made up of people of white ethnicity (90.2%), with 'other' responses for European (n = 5) and Jewish (n = 1). Almost three quarters (71.5%) of participants considered themselves Disabled by a co-occurring condition, with over half of participants (53.4%) experiencing "a little impact" on their daily activities, with a further quarter (24.4%) experiencing "a lot" of impact. The sample was highly educated, with over half of participants (62.2%) educated at undergraduate degree level or above, with "other" responses mostly focused on postgraduate courses. Almost three quarters of participants (74.1%) were from England.

Masking, which we described as: "*a term used to describe when Autistic people change their behaviour, such as reducing stimming, to make it more accepted by neurotypical people. It is also sometimes known as camouflaging*", was reported to be frequent in daily life, with 64.8% (n = 125) people masking always or most of the time, another 18.7% (n = 36) of participants reported masking about half of the time; only 2.1% (n = 4) participants reported that they never masked. Participants were asked if the support they currently received for being Autistic and any co-occurring conditions was enough, with 24 (12.5%) agreeing or strongly agreeing that the support was sufficient, and 139 (72.1%) disagreeing or strongly disagreeing that it was sufficient.

## Barriers to accessing healthcare

Participants answered questions regarding their challenges in relation to accessing healthcare, on a five-point Likert scale (Table 2). Responses showed that 13 of the 15 challenges were experienced in a healthcare context the majority of the time for over half of respondents. Furthermore, over 80% of participants experienced the challenge most of the time in relation to five items: delaying making appointments when experiencing symptoms, feeling anxious about booking appointments, delaying making healthcare appointments, feeling anxious during appointments and masking during appointments. Only two items were experienced half of the time or less by <50% of the sample: miscommunication and frustration when communicating with health professionals (43.7%) and struggling to follow post appointment instructions (35.1%).

We undertook Kruskall-Wallis tests to consider variations in healthcare experiences by diagnostic status (Table 3). The majority of these were not significant, however we identified significant associations between:

- Awareness of pain, injury or discomfort was lower in those with a formal diagnosis compared to those self-identifying [H(2) = 9.173 p = 0.010].

- Those who were formally diagnosed reported increased anxiety from sensory experiences in waiting rooms compared to those self-identifying [H(2) = 9.484 p = 0.009].

- Those who were formally diagnosed [H(2) = 14.126 p<0.001] or undergoing diagnosis [H(2) = 14.126 p<0.001] reported increased anxiety from the presence of other patients in waiting rooms compared to those who were self-identifying.

- Those who were formally diagnosed [H(2) = 8.604 p = 0.014] or undergoing diagnosis [H(2) = 8.604 p = 0.014] reported increased experiencing frustration or misunderstandings when communicating with health professionals in their preferred language compared to those who were self-identifying.

- There was a significant difference between those who were undergoing diagnosis and those who were self-identifying as Autistic in relation to anxiety due to healthcare appointments [H(2) = 8.078 p = 0.018].

- Those who were formally diagnosed were less confident that they will be understood when describing symptoms compared to those who self-identify [H(2) = 0.021 p = 0.018].

- Those who were formally diagnosed [H(2) = 11.715 p = 0.003] or undergoing diagnosis [H(2) = 11.715 p = 0.003] reported increased difficulty describing pain compared to those who were self-identifying.

- Those who were formally diagnosed [H(2) = 12.468 p = 0.002] or undergoing diagnosis [H(2) = 12.468 p = 0.002] reported increased difficulty with sensory experiences within healthcare appointments compared to those self-identifying.

Overall, these results indicate that those who are formally diagnosed or who were undergoing diagnosis at the time of data collection experienced some difficulties in accessing healthcare to a greater extent than those who were self-identifying at the time of data collection.

## Views and experiences of Autism Health Passports

The majority (65%) of participants had not heard of AHPs prior to taking part in this survey, with a further quarter only knowing 'a little' about AHPs. Only 4% of participants reported

**Table 3. Associations between frequency ("always" to "never") of reported experiences and diagnostic status.**

| Variable (reported experience) | Independent-samples Kruskall-Wallis | | Post-hoc testing (pairwise comparisons) | |
|---|---|---|---|---|
| | Test statistic, H(df) = | Significance, P = | Diagnostic status comparisons | [1]Adjusted Significance, P = |
| When I experience pain, injury, or discomfort I am _______ aware of it | 9.173(2) | 0.010 | 'Self-identify' vs 'undergoing diagnosis' | 0.53 |
| | | | 'Self-identify' vs 'formally diagnosed' | 0.011* |
| | | | 'Undergoing diagnosis' vs 'formally diagnosed' | 1.000 |
| When I experience recurrent symptoms or problems that may be intermittent, such as occasional pain, my decision and efforts to seek diagnostic healthcare appointments are _______ delayed | 0.894(2) | 0.639 (ns) | 'Self-identify' vs 'undergoing diagnosis' | 1.000 |
| | | | 'Self-identify' vs 'formally diagnosed' | 1.000 |
| | | | 'Undergoing diagnosis' vs 'formally diagnosed' | 1.000 |
| When I have to telephone a healthcare service, for example to book an appointment, I _______ feel anxious | 3.801(2) | 0.150 (ns) | 'Self-identify' vs 'undergoing diagnosis' | 0.167 |
| | | | 'Self-identify' vs 'formally diagnosed' | 1.000 |
| | | | 'Undergoing diagnosis' vs 'formally diagnosed' | 0.433 |
| When I have to telephone a healthcare service, for example to book an appointment, I _______ delay making the telephone call | 2.696(2) | 0.260 (ns) | 'Self-identify' vs 'undergoing diagnosis' | 0.438 |
| | | | 'Self-identify' vs 'formally diagnosed' | 0.436 |
| | | | 'Undergoing diagnosis' vs 'formally diagnosed' | 1.000 |
| Sensory experiences in healthcare service waiting rooms _______ make me anxious | 9.484(2) | 0.009 | 'Self-identify' vs 'undergoing diagnosis' | 0.176 |
| | | | 'Self-identify' vs 'formally diagnosed' | 0.006* |
| | | | 'Undergoing diagnosis' vs 'formally diagnosed' | 1.000 |
| The presence of other patients in healthcare service waiting rooms _______ makes me anxious | 14.126(2) | <0.001 | 'Self-identify' vs 'undergoing diagnosis' | 0.013* |
| | | | 'Self-identify' vs 'formally diagnosed' | 0.001* |
| | | | 'Undergoing diagnosis' vs 'formally diagnosed' | 1.000 |
| When communicating with healthcare professionals using my preferred type of communication (e.g.: speaking, sign language, AAC), I _______ experience frustration or misunderstandings | 8.604(2) | 0.014 | 'Self-identify' vs 'undergoing diagnosis' | 0.034* |
| | | | 'Self-identify' vs 'formally diagnosed' | 0.023* |
| | | | 'Undergoing diagnosis' vs 'formally diagnosed' | 1.000 |
| Healthcare appointments _______ make me anxious | 8.078(2) | 0.018 | 'Self-identify' vs 'undergoing diagnosis' | 0.021* |
| | | | 'Self-identify' vs 'formally diagnosed' | 1.000 |
| | | | 'Undergoing diagnosis' vs 'formally diagnosed' | 0.056 |

*(Continued)*

**Table 3.** (Continued)

| Variable (reported experience) | Independent-samples Kruskall-Wallis | | Post-hoc testing (pairwise comparisons) | |
| --- | --- | --- | --- | --- |
| | Test statistic, H(df) = | Significance, P = | Diagnostic status comparisons | [1]Adjusted Significance, P = |
| In healthcare appointments I _______ mask my Autistic communication style or behaviours | 5.384(2) | 0.068 (ns) | 'Self-identify' vs 'undergoing diagnosis' | 1.000 |
| | | | 'Self-identify' vs 'formally diagnosed' | 0.485 |
| | | | 'Undergoing diagnosis' vs 'formally diagnosed' | 0.088 |
| When I am asked to describe physical symptoms I am _______ confident I will be understood | 7.982(2) | 0.018 | 'Self-identify' vs 'undergoing diagnosis' | 0.135 |
| | | | 'Self-identify' vs 'formally diagnosed' | 0.016* |
| | | | 'Undergoing diagnosis' vs 'formally diagnosed' | 1.000 |
| When I am asked to describe pain I _______ have difficulty | 11.715(2) | 0.003 | 'Self-identify' vs 'undergoing diagnosis' | 0.002* |
| | | | 'Self-identify' vs 'formally diagnosed' | 0.048* |
| | | | 'Undergoing diagnosis' vs 'formally diagnosed' | 0.318 |
| When healthcare professionals ask me detailed questions or give me lengthy verbal instructions in a consultation I _______ find it difficult to understand | 4.531(2) | 0.104 (ns) | 'Self-identify' vs 'undergoing diagnosis' | 0.166 |
| | | | 'Self-identify' vs 'formally diagnosed' | 0.183 |
| | | | 'Undergoing diagnosis' vs 'formally diagnosed' | 1.000 |
| When I am emotional or distressed in a healthcare consultation or setting, my communication skills are _______ reduced | 6.030(2) | 0.049 | 'Self-identify' vs 'undergoing diagnosis' | 0.055 |
| | | | 'Self-identify' vs 'formally diagnosed' | 0.164 |
| | | | 'Undergoing diagnosis' vs 'formally diagnosed' | 1.000 |
| Sensory experiences within healthcare appointments are _______ difficult for me | 12.468(2) | 0.002 | 'Self-identify' vs 'undergoing diagnosis' | 0.029* |
| | | | 'Self-identify' vs 'formally diagnosed' | 0.002* |
| | | | 'Undergoing diagnosis' vs 'formally diagnosed' | 1.000 |
| I find that lengthy instructions, such as when to fill prescriptions, have tests, and make follow-up appointments, are _______ easy to understand | 1.867(2) | 0.393 (ns) | 'Self-identify' vs 'undergoing diagnosis' | 1.000 |
| | | | 'Self-identify' vs 'formally diagnosed' | 0.525 |
| | | | 'Undergoing diagnosis' vs 'formally diagnosed' | 1.000 |
| With regard to adhering to post-appointment instructions (such as wound care or medications) I manage to _______ follow instructions exactly | 1.805(2) | 0.405 (ns) | 'Self-identify' vs 'undergoing diagnosis' | 1.000 |
| | | | 'Self-identify' vs 'formally diagnosed' | 0.602 |
| | | | 'Undergoing diagnosis' vs 'formally diagnosed' | 1.000 |

[1]Adjusted via Bonferroni correction for multiple tests

*Significant association

**Table 4. Views of Autism Health Passports.**

| I feel that a completed Autism Health Passport would... | Strongly agree/ Agree | | Neither agree nor disagree | | Strongly disagree/ disagree | | Prefer not to say |
|---|---|---|---|---|---|---|---|
| | N | % | N | % | N | % | |
| be helpful in communicating enough information about my specific Autistic presentation. (n = 182) | 106 | 58.3% | 38 | 20.9% | 37 | 20.3% | 1 (0.5%) |
| have the ability to adequately inform medical professionals about my co-occurring conditions (e.g., epilepsy, EDS). (n = 182) | 106 | 58.3% | 43 | 23.6% | 29 | 15.9% | 4 (2.2%) |
| accurately convey my overall health and well-being. (n = 181) | 77 | 42.6% | 45 | 24.9% | 57 | 31.5% | 2 (1.1%) |
| reduce the need for me to give the same information to different members of staff? (*E.g., GP to midwife, midwife to Health Visitor*). (n = 181) | 124 | 68.5% | 26 | 14.4% | 29 | 16.0% | 2 (1.1%) |
| help me receive the same care quality as a non-Autistic person. (n = 180) | 50 | 27.7% | 38 | 21.1% | 90 | 50% | 2 (1.1%) |
| be useful in routine appointments. (n = 181) | 104 | 57.5% | 42 | 23.2% | 31 | 17.1% | 4 (2.2%) |
| be useful in emergency medical situations. (n = 180) | 140 | 77.8% | 22 | 12.2% | 17 | 9.4% | 1 (0.6%) |
| be useful to effectively communicate my needs in-between appointments *e.g.: when booking appointments).* (n = 178) | 85 | 47.8% | 33 | 18.5% | 58 | 32.6% | 2 (1.1%) |

knowing 'a lot' about AHPs. When asked if they had ever seen an AHP prior to the survey, almost three quarters (72%) noted that they had not. Unsurprisingly, only 3 participants (1.5%) used an AHP with health professionals 'most of the time' or 'about half of the time', with 90% 'never' using an AHP.

Participants responded to a range of Likert scale questions on the potential utility of AHPs (Table 4). Overall, around half of participants agreed or strongly agreed that AHPs could communicate information about their Autistic presentation (58.3%), co-occurring conditions (58.3%), and their overall wellbeing (42.6%), which two thirds of participants thought could reduce the need to repeat information to multiple health professionals (68.5%). Participants agreed or strongly agreed that this could be useful between appointments, such as when booking appointments, (47.8%), during routine appointments (57.5%) and particularly during emergency care (77.8%). However, only one quarter of participants (27.7%) agreed or strongly agreed that an AHP would allow for equal treatment compared to neurotypical peers, with half of participants (50.0%) disagreeing or strongly disagreeing with this statement.

179 participants completed at least six of the AHP opinion questions (Table 4). We combined these and divided by the number of questions answered to create a mean AHP score. The Minimum score available was 1, and the maximum score available was 5, with a higher score demonstrating stronger *disagreement* that AHPs could improve healthcare accessibility for Autistic people. We considered variations in AHP score by diagnostic status. The mean AHP score for all three groups was between 2 ('somewhat agree') and 3 ('neither agree nor disagree'). The AHP score was highest in those who were diagnosed (n = 100, M = 2.66 SD = 0.91), with those self-identifying slightly lower (n = 40 M = 2.57, SD = 0.96). Those who were undergoing diagnosis had the lowest AHP score (n = 39, M = 2.25, SD = 0.78). A one way ANOVA did not demonstrate that AHP score was significant between groups [$F_{(2,176)}$ = 2.885, p = .058].

## Context: Health professionals discriminate against Autistic people

Despite responses to the closed questions appearing slightly positive about the usefulness of AHPs, closed text responses contained many barriers to use. The most common concept, identified in 65 participants' responses, was health professionals' lack of understanding of Autism and the resulting discrimination towards Autistic patients. This lack of understanding was attributed to a lack of training (n = 22) and was contextualised by nine participants reporting

the routine 'stigmatisation' of Autistic people in society. Participants reported specific concerns related to staff using outdated "stereotypes" or "making assumptions" about Autism, such as thinking the person was "being thick or a bit rubbish" causing them to "not (be) taken seriously", and infantalised or patronised by staff. Some participants (n = 13) drew on their previous negative experiences of interacting with health professionals where disclosing their diagnosis resulted in different and undesired treatment. These negative experiences, or fear of them, resulted in a decision to not directly "out" themselves as Autistic to health professionals. For example:

> Honestly, the ongoing stigma of autism by many in healthcare settings—it would make me anxious if a doctor knew I was autistic as I believe their pre-conceived ideas of autism could override how they treat me. If healthcare authorities can be trained to fully understand neurodivergent conditions then I can see this passport being helpful, but the training would be much more helpful to autistic people who (can) then feel safe to openly share their diagnosis and be themselves. (Formally diagnosed, primarily speaking white woman aged 40).

Alongside lack of diagnosis sharing, some participants (n = 12) explicitly mentioned masking their Autism during appointments to reduce discrimination and receive better treatment. This was instead of disclosing Autism and asking for accommodations, as accommodation requests often lead to no change in treatment or discrimination:

> I sometimes find it easier to not mention being autistic and to work hard trying to mask, as it costs energy to get into discussions about my autism presentation, and if I'm seeking medical help, I'm probably close to sensory overwhelm in any case. Sometimes I feel as though saying 'I'm autistic' doesn't make any difference to my treatment anyway, so it's easier to not disclose. (Formally diagnosed, primarily speaking white woman aged 35).

### Experiences of using AHPs and trying to communicate Autism-related needs

Nine participants reported their experience of using an AHP in open text questions. The most reported experience among this small group was having their AHP ignored by a health professional (n = 4). In addition, two participants who had not used an AHP created their own written summary, but also found that it was ignored when presented to a health professional:

> I think that they are only good if healthcare professionals read them. I attached a section about all my medical needs to my maternity notes but the midwife in the hospital didn't look at it. (Formally diagnosed, primarily speaking white woman aged 36).

Of these nine participants, none reported *only* a positive experience, with two participants neutrally reporting that they use the tool, such as: "I use it when consulting GP". A further two participants reported conflicting positive and negative experiences: "I had an autism passport for my maternity care. It was somewhat helpful but also slightly disregarded by some." Alongside this, three participants had completed AHPs but not used them yet, with two stating that this was due to concerns about being stigmatised because of their disclosure: "I have never remembered to take my own with me and I am afraid of being treated badly if I show it." Furthermore, two participants who did not report using an AHP stated their resistance to using a tool created by the National Autistic Society, preferring to "use one led and made by autistic people" instead.

Nine participants noted that they had created their own written document to share with health professionals or to use within health appointments. These were heterogeneous, including a one page "about me" document, a list of questions to ask, or a written summary of the problem they were experiencing, including stating that they had left a written copy with the doctor to have on file:

> I write down everything I want the doctor to know about or that I want to ask them. It helps to have it written down so they can get a clear idea of how this issue is effecting you. Plus it makes their notes after more detailed especially if you leave the paper with them. Therefore getting you more helpful and better equipped healthcare. (Formally diagnosed, primarily speaking white woman aged 26).

A further nine participants noted that they used other accessibility aids, including: "stickman communication cards", "sunflower lanyard", "Autistic research passport", "COVID info card from Autism Understanding Scotland", and "alert card issued by police". Having a trusted person, including a family member or carer, as an interpreter or advocate with them in the appointment was reported by eight participants as an accessibility measure used to receive better care:

> I think it would be helpful to be able to have someone who can relay information as a sort of interpreter between myself and healthcare professionals. I often use my husband for this role. (Formally diagnosed, primarily speaking white gender fluid person aged 30).

> The only tool ever effective is borrowing a white cis man to repeat what I say. (Formally diagnosed, preferred communication by typing, white woman aged 48).

### Utility of Autism Health Passports

Within the open text question focused on barriers to using AHPs, 12 participants noted that health professionals wouldn't know what an AHP was, and 58 participants were concerned that the AHP would be ignored by health professionals:

> I think a lot of health professionals probably wouldn't read it beforehand unless it became part of the mandatory process to read before appointments. (Formally diagnosed, primarily speaking white woman aged 21).

When describing how they thought the AHP would affect their care, many used negative words, including "prejudice" (n = 4), "infantilising" (n = 3) and "misunderstood" (n = 2), highlighting that some participants thought AHPs could exacerbate Autism discrimination, for example:

> I find disclosure of my autism very challenging in healthcare settings. Some medics deny the diagnosis (saying:) "you can't be", whilst others instantly treat me like I'm stupid. I have to be very careful about disclosure and try and judge it in each individual case. I'm not sure that something on file would help me as I would lose this flexibility. (Formally diagnosed, primarily communicates in writing, white woman aged 43)

Further barriers identified focused on the utility of AHPs. For seven participants with complex disabilities, co-occurring conditions or conditions that vary depending on external

circumstances, the passport was seen as too simple and not able to convey all of the necessary information: "My passport would be huge as I have a lot of co-occurring conditions". Participants (n = 4) also noted that it was unclear how the AHP would be used for online or virtual appointments. A further barrier for those who self-identified as Autistic (n = 5) was being unsure if an AHP would be accepted by a health professional without a formal diagnosis.

A minority of participants (n = 7) who hadn't previously used an AHP responded to the question on barriers to AHP use by reporting potential benefits in using AHPs and said that they would consider using one in the future, or that they would recommend one to their Autistic children:

> It could be very useful for me and might help me to be understood more. (Formally diagnosed, primarily speaking mixed race woman aged 30).

> While it's not necessarily the life-saving aid for me, I will look into getting one of these for my teenage son when he needs to attend appointments without me in a few years. He gets less verbal under stress and forgets details in a pressurised situation like a doctor appointment. This would be ideal for him. (Formally diagnosed, primarily speaking white woman aged 47).

Participants provided suggestions for improving healthcare access, including additional training and education to health professionals about Autism, longer appointment times and having an Autism support worker present in the medical office.

## Discussion

Our participants frequently reported challenges accessing healthcare, as has been established in previous research [5]. When considering diagnostic status against these healthcare access challenges, we identified many non-significant results. This could be considered as evidence that those without a formal Autism diagnosis may face similar healthcare access challenges to their diagnosed peers. However, it did appear that those undergoing diagnosis at the time of data collection were more similar to diagnosed participants than to those who self-identified as Autistic, particularly relating to anxiety regarding other patients in waiting rooms, miscommunication and frustration when communicating with health professionals and difficulty describing pain. This association could be related to the known difficulties of securing an Autism diagnosis through health services [30], which may present additional challenges for Autistic women [31], or to a process of becoming more aware of Autistic traits and "demasking" following diagnosis [32].

It is important that healthcare interventions are designed to take into account the needs of populations they target, that they can feasibly be implemented into healthcare systems, and that they are theoretically robust [33]. Arguably, it would be beyond the scope of even the most excellent health passport to address the barriers to accessing healthcare that our participants reported, including anxiety about and delays when booking appointments; anxiety and masking during appointments and reduced communication skills when distressed. Furthermore, the challenges of accessing different aspects of healthcare can vary, from the feeling of ultrasound jelly on the abdomen during maternity care [17] to the noise of an Magnetic Resonance Imaging machine [14], requiring tailored solutions. These challenges should be considered when designing new evidence-based interventions which aim to reduce health inequalities for Autistic people. In our realist review of Autism Health Passports, we identified barriers to AHP use at the interpersonal, environmental and societal levels [27], and these were also identified in survey responses. Thus, we recommend that all three of these levels are considered in any intervention which proposes to utilise an AHP or similar tool. The Autistic

SPACE framework, developed by Autistic doctors, may hold promise [34] but has not yet been evaluated for feasibility of implementation.

Our survey was the first that we know of that asked Autistic people about their views and experiences of AHPs, outside of developing or evaluating such passports. AHPs were not widely known about or used by our participants, who reported multiple Autism-related barriers to accessing healthcare. Whilst participants' responses to closed questions reported they felt that AHPs could be slightly useful (between "agree" and "neither agree or disagree" on a 5-point Likert scale) at reducing health inequalities, the major finding of our qualitative content analysis was that health professionals' lack of understanding of Autism, as is well established [6], is a barrier to sharing a diagnosis or requesting adjustments due to fear of receiving worse treatment. In maternity settings, this goes further as declarations can lead to inappropriate social work intervention [35] or unfounded suggestions of Fabricated and Induced Illness [36]. Furthermore, participants felt that AHPs would be likely to be ignored by health professionals, highlighting another systemic barrier to their use.

Whilst policy and clinical guidance [24] recommend AHPs as an equality measure, our realist review of the evidence [27] and the results of this survey suggest that in *isolation* and within a healthcare context that routinely fails to meet the needs of Autistic patients, Autistic people believe that AHPs can achieve very little. They also put the onus on Autistic patients to present information to their clinician. This can be challenging for Autistic people because of known communication differences which make healthcare appointments a source of anxiety, and known challenges with executive functioning, which would be required to remember to bring a paper-based health passport to appointments and present it to the clinician.

To our knowledge, AHP-type tools are not embedded into routine care for Autistic adults anywhere in the UK. However, at St George's University Hospitals NHS Foundation Trust in London, UK, 95% of patients with intellectual disabilities use a hospital passport or similar tool [37], showing that it is theoretically possible to embed these tools into routine practice. However, in other examples, even when prompts to use accessibility tools for Autistic patients, such as AHPs, are embedded into electronic systems, shortage of clinician time and clinician lack of confidence results in these reminders being dismissed without action more than half of the time [25]. It is therefore of little surprise that the few participants in our study who had tried to use an AHP in their healthcare consultations reported mixed or negative staff responses.

The systemic barriers to the use of AHPs are not isolated to supporting Autistic patients; the UK National Review of Asthma Deaths identified that 70% of the children who died did not have a written asthma management plan, despite the known mortality risk in childhood asthma [38]. Accordingly, before recommending them as a tool that *will* reduce health inequality, healthcare systems must be redesigned in such a way that these health passport type tools become a valuable resource, rather than a tokenistic low-cost intervention. Improved education and training for healthcare providers on Autism through a neurodiversity affirming lens and how to communicate with Autistic people was stated as a desired outcome from many participants, and this is something that we wholeheartedly endorse as one *part* of creating a cultural change. Making healthcare staff more accepting of Autistic people could reduce the reliance on AHPs. Furthermore, making healthcare a more supportive environment for Autistic health professionals to "out" themselves, without fear of negative consequences, will also contribute to culture change where Autism is better understood by health professionals.

## Strengths and limitations

193 Autistic people who had been pregnant from the UK took part in our survey. To date, Autistic women and other adults AFAB have been under-diagnosed and under-researched

[39]. Our research reinforces that Autistic women and others AFAB are interested in taking part in well designed, neurodiversity affirming research studies. Our study was also the first attempt to understand the utility of health passports for Autistic people outside of a study which designed and evaluated such a tool, adding to the literature. Furthermore, meaningful Autistic representation on our research team is an epistemically important strength [40]. That said, we did not undertake piloting with field experts or run internal consistency tests, meaning that we cannot guarantee the validity or reliability of the instrument. Furthermore, our research did not utilise pre-existing measures which had been validated in research with Autistic people, which may have resulted in issues relating to construct validity [41]. Although, we used measures of healthcare accessibility that had been previously used in research with Autistic adults [42], increasing the likelihood that these measures were valid. The use of an online survey may have reduced social desirability bias in answers.

The generalisability of our findings is likely to be limited by self-selection bias, resulting in our participants being a relatively privileged sample, including >60% of participants educated to at least undergraduate degree level, >90% identifying as having white ethnicity and 85% of participants were primarily speaking. Accordingly, those who are additionally marginalised in relation to demographics including ethnicity, education and poverty may find AHPs even more challenging to use [43]. Furthermore, the time needed to complete our survey makes it unlikely that many Autistic people with intellectual disabilities would have completed the full survey. Additionally, we did not specifically ask participants if they had an intellectual disability.

## Conclusion

At present it is not clear what value, if any, Autism Health Passports have in healthcare, including maternity services. Policies that recommend AHPs appear to disregard the multiple systemic barriers to their use, including a culture where being Autistic is only seen as a deficit, and a lack of clinician time and training to make use of any AHPs presented. Further research is urgently required to identify ways in which healthcare systems can be made more accessible to Autistic people. This should be strongly grounded in known barriers to healthcare use and additional known challenges that Autistic people face in other contexts. Furthermore, such an intervention requires a strong theoretical basis, including a theory of change which takes account of known systemic barriers. Only then can such interventions reduce health inequality for Autistic people.

## Supporting information

**S1 Appendix. Survey instrument.**
(DOCX)

## Acknowledgments

We gratefully acknowledge Carol McIntyre for data cleaning and Libby Foot for proof-reading assistance.

## Author Contributions

**Conceptualization:** Aimee Grant, Kathryn Williams, Hayley Morgan, Rebecca Ellis, Amy Brown.

**Data curation:** Aimee Grant, Sebastian C. K. Shaw.

**Formal analysis:** Aimee Grant, Sarah Turner, Sebastian C. K. Shaw.

**Funding acquisition:** Aimee Grant, Kathryn Williams, Amy Brown.

**Methodology:** Aimee Grant, Kathryn Williams, Hayley Morgan, Rebecca Ellis, Amy Brown.

**Project administration:** Aimee Grant.

**Supervision:** Aimee Grant, Amy Brown.

**Validation:** Aimee Grant.

**Visualization:** Aimee Grant, Sebastian C. K. Shaw, Amy Brown.

**Writing – original draft:** Aimee Grant, Sarah Turner.

**Writing – review & editing:** Aimee Grant, Sarah Turner, Sebastian C. K. Shaw, Kathryn Williams, Hayley Morgan, Rebecca Ellis, Amy Brown.

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
