## [Decision Letter · Decision Letter 0]

6 Mar 2024

PONE-D-23-26665“I am afraid of being treated badly if I show it”: a cross-sectional study of healthcare accessibility and Autism Health Passports among UK Autistic adultsPLOS ONE

Dear Dr. Grant,

Thank you for submitting your manuscript to PLOS ONE. After careful consideration, we feel that it has merit but does not fully meet PLOS ONE’s publication criteria as it currently stands. Therefore, we invite you to submit a revised version of the manuscript that addresses the points raised during the review process.

I agree with the reviewer that this manuscript has potential but could benefit from significant revision. In addition to addressing the points noted by the reviewer, please also introduce the nature and function of Autism Health Passports in the Introduction. This was vital context that was missing from the current version of the manuscript.

We look forward to receiving your revised manuscript.

Kind regards,

Emily Lund

Academic Editor

PLOS ONE

Journal Requirements:

“SS is the research lead for Autistic Doctors International. KW is the research director for Autistic UK. The remaining authors declare no competing interests.”

We note that one or more of the authors are employed by a commercial company: Autistic UK & Autistic Doctors International

A.        Please provide an amended Funding Statement declaring this commercial affiliation, as well as a statement regarding the Role of Funders in your study. If the funding organization did not play a role in the study design, data collection and analysis, decision to publish, or preparation of the manuscript and only provided financial support in the form of authors' salaries and/or research materials, please review your statements relating to the author contributions, and ensure you have specifically and accurately indicated the role(s) that these authors had in your study. You can update author roles in the Author Contributions section of the online submission form.

B.  Please also provide an updated Competing Interests Statement declaring this commercial affiliation along with any other relevant declarations relating to employment, consultancy, patents, products in development, or marketed products, etc. 

Additional Editor Comments (if provided):

I agree with the reviewer that this manuscript has potential but could benefit from significant revision. In addition to addressing the points noted by the reviewer, please also introduce the nature and function of Autism Health Passports in the Introduction. This was vital context that was missing from the current version of the manuscript.

Reviewers' comments:

Reviewer's Responses to Questions

**Comments to the Author**

1. Is the manuscript technically sound, and do the data support the conclusions?

Reviewer #1: Yes

2. Has the statistical analysis been performed appropriately and rigorously? 

Reviewer #1: Yes

3. Have the authors made all data underlying the findings in their manuscript fully available?

Reviewer #1: No

4. Is the manuscript presented in an intelligible fashion and written in standard English?

Reviewer #1: Yes

5. Review Comments to the Author

Reviewer #1: Thank you very much for the opportunity to review this interesting study. Your outcomes are important to the autistic community. Please see below some comments to improve the manuscript.

Background:

A generally well-written section.

Lines 65-66: I do not think that you can generalise this here. I agree that the experiences of autistic adults in healthcare need more reserach, but, I do not agree to cite a paper published 10 years ago (2014) and say that their experiences are not explored. For instance, you can find some great job done by medical imaging professionals around autistic people experiences in MRI, which I suggest citing (preferably in the discussion of your manuscript) to show the wider context of research aorud this topic.

http://dx.doi.org/10.1089/aut.2022.0051

https://doi.org/10.1177/13623613211065542

https://doi.org/10.1186/s12913-023-10333-w

Line 79: You state ''numerous documents'' but you only cite one. Please consider adding a couple of referneces here to support your statement.

Community involvement and reflexivity:

I would suggest expanding a bit more regarding reflexivity. You can report gender, age, highest qualifications, ethnicity of the team, to allow the readers get a better idea of your views, experiences, and perceptions on the topic.

Survey design:

I can see that you have not performed piloting with field experts, and also that you did not run any internal consistency tests (e.g. Cronbach's alpha). This means that you have to explicitly state in your limitations that you cannot quarantee either the validity or the reliability of the instrument.

Participants and eligibility criteria:

Line 129: I think you cannot state that multiple responses from the same participant were prevented by Qualtrics, since this survey software only prevents people to answer the survey twice from the same IP address. There is always the potentail risk of some people answering more than once from a different IP. Please change this sentence.

Lines 138-139: Did you inform the participants beforehand regarding the incentives given after the data collection? Was this written on the information sheet?

It seems that you collected some personal data to facilitate entry into the prize draw. Were the participants informed about this before entering the survey? This is important, since the anonymity of the respondents was not ensured. Was it a voluntary process, or were all the participants required to put their personal data in?

Analysis:

Please consider providing the readers with the full terms for SPSS and ANOVA.

I can see that you chose to analyse all data from respondents who had anwered at least their demographics. This is a correct strategy. However, was any attrition in your survey? Did you notice any respondents leaving the survey in different stages before completing it? If yes, you have to report attrition bias, and also to clarify if your quantitative analyses have taken the actual numbers of responses for each question, or the total number of responses, into account.

Lines 156-157: You say here that you performed a thematic analysis. However, I cannot see any clear themes that derived from this in the results section. Thematic analysis is about identifying themes from the data. It seems that you performed a conceptual content analysis instead, since I can see reporting frequencies in the results of the open-ended questions. If this is the case, I suggest changing the above statement (and the ref as well), replacing this with content analysis, and adding a clarification about your strategy (e.g. how did you perform coding of data, how did you group them into common categories, etc.). In the results, you can always report the frequencies in which each category appears, and of course your representative quotes as you already did.

Results:

Please consider changing your stategy regarding analysis and presentation of the qualitative data, as above.

Table 3: There is no need to put all these statistics into Table 3, since this turns out to be confusing to the reader. I suggest either depicting only the statistically significant results of the analyses, or taking the entire table out and leaving the text above which summarises well all significant results.

Discussion:

A well-balanced discussion summarising the key findings of the study. Perhaps consider only enriching this by adding some refs (like the ones suggested in the background) from different disciplines to show to highlight the problem in the light of the wider literature.

Limitations:

This is fair enough, provided that you will add the limitations suggested above.

6. PLOS authors have the option to publish the peer review history of their article (what does this mean?). If published, this will include your full peer review and any attached files.

Reviewer #1: No

---

## [Author Response · Author response to Decision Letter 0]

30 Apr 2024

Please see word document uploaded as "Response to reviewers"

---

## [Editor Report · Decision Letter 1]

2 May 2024

“I am afraid of being treated badly if I show it”: a cross-sectional study of healthcare accessibility and Autism Health Passports among UK Autistic adults

PONE-D-23-26665R1

Dear Dr. Grant,

We’re pleased to inform you that your manuscript has been judged scientifically suitable for publication and will be formally accepted for publication once it meets all outstanding technical requirements.

Kind regards,

Emily Lund

Academic Editor

PLOS ONE
---

## [Editor Report · Acceptance letter]

7 May 2024

PONE-D-23-26665R1 

PLOS ONE

Dear Dr. Grant, 

I'm pleased to inform you that your manuscript has been deemed suitable for publication in PLOS ONE. Congratulations! Your manuscript is now being handed over to our production team.

Kind regards, 

on behalf of

Dr. Emily Lund 

Academic Editor

PLOS ONE